# Encrypt with Your Mind: Reliable and Revocable Brain Biometrics via Multidimensional Gaussian Fitted Bit Allocation

**DOI:** 10.3390/bioengineering10080912

**Published:** 2023-08-01

**Authors:** Ming Li, Yu Qi, Gang Pan

**Affiliations:** 1State Key Lab of Brain-Machine Intelligence, Hangzhou 310018, China; 2College of Computer Science and Technology, Zhejiang University, Hangzhou 310027, China; 3Affiliated Mental Health Center & Hangzhou Seventh Peoples Hospital, MOE Frontier Science Center for Brain Science and Brain-Machine Integration, Zhejiang University School of Medicine, Hangzhou 310030, China

**Keywords:** authentication, biometrics, brain decoding, electroencephalogram, intracortical brain signals, key generation, local field potential

## Abstract

Biometric features, e.g., fingerprints, the iris, and the face, have been widely used to authenticate individuals. However, most biometrics are not cancellable, i.e., once these biometric features are cloned or stolen, they cannot be replaced easily. Unlike traditional biometrics, brain biometrics are extremely difficult to clone or forge due to the natural randomness across different individuals, which makes them an ideal option for identity authentication. Most existing brain biometrics are based on electroencephalogram (EEG), which is usually demonstrated unstable performance due to the low signal-to-noise ratio (SNR). For the first time, we propose the use of intracortical brain signals, which have higher resolution and SNR, to realize the construction of the high-performance brain biometrics. Specifically, we put forward a novel brain-based key generation approach called multidimensional Gaussian fitted bit allocation (MGFBA). The proposed MGFBA method extracts keys from the local field potential of ten rats with high reliability and high entropy. We found that with the proposed MGFBA, the average effective key length of the brain biometrics was 938 bits, while achieving high authentication accuracy of 88.1% at a false acceptance rate of 1.9%, which is significantly improved compared to conventional EEG-based approaches. In addition, the proposed MGFBA-based keys can be conveniently revoked using different motor behaviors with high entropy. Experimental results demonstrate the potential of using intracortical brain signals for reliable authentication and other security applications.

## 1. Introduction

Identity authentication plays an important role in the information security field, particularly for applications that deal with sensitive or private information (e.g., financial, medical, military, and shopping information). Passwords and personal identification numbers (PINs) are frequently used for access authentication. However, users usually use short passwords or PINs that are easy to remember, but long and complicated passwords are required to realize sufficient security. In addition, users often set the same password across different applications; thus, once the passwords are stolen, guessed, or hacked, the hacker can access multiple applications fraudulently [1].

Biometric features, e.g., fingerprints, the face, and the iris, have been a popular alternative for identity authentication [2]. Biometric features are natural biological components of the human body; thus, users do not have to remember long passwords. Due to the biological uniqueness of individuals, biometric features contain rich information to guarantee authentication security [3]. Thus, biometrics are considered an ideal way to validate authorized users [3]. In the following, we survey the existing biometrics and divide them into two categories, i.e., irrevocable and revocable biometrics. **(1) Irrevocable biometrics.** Irrevocable biometric features contain immutable patterns, and representative examples are commonly used, e.g., fingerprints [4,5], face [6] and the iris [7]. Several studies have investigated such biometrics in recent decades; however, they have a critical limitation, i.e., they are irrevocable and can be copied easily by attackers [8,9,10,11]. For example, fingerprints left on keyboards or cups can be collected easily. In addition, faces are vulnerable to camouflage from printed photos. To address this problem, some studies have proposed using internal biological properties as biometric features, including DNA [12] and electrocardiogram (ECG) signals [13,14]). Ref. [12] employed DNA binary strands to provide rapid encryption and decryption using information hiding for steganography. Ref. [14] adopted ECG signals to generate cryptographic keys, and they achieved 100-bit keys with 95% reliability extracted from ECG signals acquired over multiple sessions. Such internal biometrics are more difficult to clone; however, they are irrevocable. In other words, once the biometric features are cloned or stolen, they cannot be substituted easily. **(2) Revocable biometrics.** Compared to irrevocable biometrics, revocable biometrics can be updated once stolen. The most commonly used revocable biometrics are behavioral biometrics. In 2002, Monrose et al. [15] proposed a behavioral biometric feature comprising a combination of duration and delay between keystrokes. In addition, handwritten signatures [16,17,18] are another example of a behavioral biometric feature, where the local features of the signature are used for application. However, the keys generated by behavioral biometrics are generally very short, e.g., approximately 40 bits [16], which may be insufficient security for applications with high security requirements. Recent studies have demonstrated that the human brain can provide superior revocable biometric features [19,20,21,22,23,24]. Brain waves are continuous physiological signals with variable characteristics that change in real time; thus, it is extremely difficult to steal or imitate brain signals. A previous study [20] represented EEG signals as a graph based on within-frequency and cross-frequency functional connectivity estimates, and they utilized a graph convolutional neural network for person identification. Another study [23] integrated transfer learning and dictionary learning in a learning model for EEG emotion recognition. However, these methods ignore the cancellability of brain biometrics.

Brain-based biometrics are potentially revocable; however, existing EEG-based methods suffer from poor performance. A major problem with EEG is its low signal-to-noise ratio (SNR) because the electrical signals in the brain decay significantly as they pass through the skull and scalp. Thus, an EEG-based system cannot achieve sufficient reliability due to the signal quality, and the revocability is highly limited [25]. In addition, the long-term stability of EEG is also unsatisfactory. A previous study [26] evaluated the performance of EEG over time and identified a significantly decreasing trend. Therefore, determining how to realize high reliability and long-term stability is an essential but challenging problem for brain-based biometrics. Compared with EEG signals, intracortical brain signals are an effective option in terms of constructing the high-performance brain biometrics, which are recorded with electrodes placed directly on the cortex such that the signals have higher resolution and SNR [27,28]. The main contributions of our work can be summarized as follows:To the best of our knowledge, this is the first study to encode intracortical brain signals into reliable long keys.In this paper, we propose a novel approach called multidimensional Gaussian fitted bit allocation (MGFBA) to encode brain signals into digitalized keys.We found that with the proposed MGFBA, the average effective key length using the intracortical brain signals of 10 rats was 938 bits, and we achieved high authentication accuracy of 88.1% at a false acceptance rate of 1.9%, which is a significant improvement over conventional EEG-based approaches.Our MGFBA-based keys can be conveniently revoked using different motor behaviors. The experimental results demonstrate the potential of using intracortical brain signals for reliable authentication and other security applications.

## 2. Methods

### 2.1. Overview

In this paper, we propose a new type of brain biometrics, which utilize intracortical brain signals to realize authentication. The proposed biometrics contains two main processes, i.e., the registration and authentication processes, as shown in Figure 1. During the registration process, information about the subjects is collected, including selecting reliable features and generating key templates for each subject. Specifically, we collect the LFP signals of each subject performing the selected behavior as input, and we then preprocess the signals and extract features. After the MGFBA quantization process, we obtain the indices of reliable features as helper data and key templates for each subject stored in a secure storage medium. If a user is to be verified, the authentication process regenerates the key with MGFBA method and compares it to the key template for identity authentication. Here, we collect the LFP signals of the user performing the selected behavior as input, and then, we compute the key with MGFBA quantization. Finally, we compare the similarity of the regenerated key and the key template to validate the user’s identity.

### 2.2. Signal Preprocessing and Feature Extraction

The raw neural signals are processed by a 0.5–300 Hz bandpass filter (i.e., a two-order Butterworth filter) to obtain the LFP signals. According to the behavior data, we separate the performing segments of the neural signals for key generation. Within the running behavior, all data are used; within the grabbing behavior, we only preserve the segments between the reach and take-back action; within the pressing behavior, only the segments with lever pressure above a certain threshold are employed. The neural signal segments are then concatenated temporally for further feature extraction. For each channel of LFP signals, we compute the power of five classical frequency bands as features, i.e., delta band (δ, 0.5–4 Hz), theta band (θ, 4–8 Hz), alpha band (α, 8–12 Hz), beta band (β, 12–30 Hz), and gamma band (γ, 30–300 Hz). Specifically, we take 2-s long neural signals as a trial and the power of five frequency bands for each trial was calculated by applying 512-point windows with overlap to provide one sample every 100 ms. Then, a Hanning window followed by a fast Fourier transform to each window was used to calculate time domain squared power. The log transform of the frequency power was computed as features as follows:(1)Power(n)=log(p(n))
where *n* stands for each frequency band and p(n) is the time domain squared power belonging to the band *n*. In addition, we append an additional feature log(∑p(n)), i.e., the log value of the sum of the frequency power of all frequency bands. In total, there are 96 features of each trial, with 16 channels considered in our experiments.

### 2.3. MGFBA Quantization

MGFBA quantization converts real-valued LFP features into binary strings for each subject. Figure 2 illustrates the specific steps of the MGFBA quantization process. Here, Step I is a preprocessing step to compute the population parameters, i.e., the population center and margins, which are computed based on the input parameter α and the population statistics. Step II involves the feature selection and key regeneration processes for each subject. Firstly, the individual statistics are taken as the primary input, and the population center and margins for each feature (computed in Step I) are then used to select the most reliable features for each subject. We then compute the code of each reliable feature and combine the codes to generate a key template for each subject. The population parameters and indices of the selected features are stored as helper data for subsequent use. Step III is an authentication step. Here, the feature indices calculated in Step II are applied to regenerate the key using the population parameters. By comparing the similarity of the regenerated key and the target key template, the authentication is successful only if the similarity is greater than a specific threshold for the given subject. Prior to discussing the proposed MGFBA in detail, we first present the assumptions considered in our experiments as follows.

**Assumption** **1.**
*Samples from the entire population are assumed to be able to represent the probability density function of each feature, and these probability density functions are considered multivariate Gaussian distributions. Here, for each LFP feature f of K dimensional space, we denote the probability density function of the population as $PDF_{pop}^{f}∼N\left(\mu_{pop}^{f},\sum_{pop}^{f}\right)$, where $\mu_{pop}^{f}\in R^{K}$, $\sum_{pop}^{f}\in R^{K\times K}$.*


**Assumption** **2.**
*The LFP features of each subject are computed during registration, and these are also considered multivariate Gaussian distributions. Similarly, for each LFP feature f of K dimensional space for subject $S_{i}$ (i∈[1,k], where k is the number of subjects), we represent the probability density function of $S_{i}$ as $PDF_{S_{i}}^{f}∼N\left(\mu_{S_{i}}^{f},\sum_{S_{i}}^{f}\right)$, where $\mu_{S_{i}}^{f}\in R^{K}$, $\sum_{S_{i}}^{f}\in R^{K\times K}$.*


For the 1-bit case, as shown in Figure 3a, the central red curve refers to the probability distribution of a feature of the entire population, and the yellow and green curves refer to the probability distributions of the same feature for $Subject_{a}$ and $Subject_{b}$, respectively. As shown in Figure 3b, for the 2-bit case, the central red circle refers to the probability distribution of the entire population, and the other four circles represent the probability distributions of different subjects in the population. In addition, the solid and dotted lines represent the population center and the boundaries of different code areas, respectively. For simplicity, we draw the center border of the project of two-dimensional probability distributions in the plane for the 2-bit case. In addition, the exact feature distribution of each rat is shown in Appendix A.

**Population Parameter Computation.** The first step of MGFBA quantization is to compute the essential parameters of the population, i.e., the center point and the margin. As mentioned previously, the features of the population are known and considered Gaussian distributions. Thus, we attempt to fit the data into multivariate Gaussian functions to obtain the value of $\mu_{pop}^{f}$ and $\sum_{pop}^{f}$. In other words, for the 1-bit case, $\mu_{pop}^{f}=\mu$ and $\sum_{pop}^{f}=\sigma$ in $f\left(x\right)=\frac{1}{\sqrt{2\pi }\sigma }e^{-\frac{{\left(x-\mu \right)}^{2}}{2\sigma^{2}}}$. Similarly, for the 2-bit case, $\mu_{pop}^{f}=\left(\mu_{X},\mu_{Y}\right)$ and $\sum_{pop}^{f}=\left[\sigma_{X},0;0,\sigma_{Y}\right]$, with ρ=0 for ease of calculation, in $f\left(x,y\right)=\frac{1}{2\pi \sigma_{X}\sigma_{Y}\sqrt{1-\rho^{2}}}\times e^{-\frac{1}{2(1-\rho^{2})}\left[{\left(\frac{x-\mu_{X}}{\sigma_{X}}\right)}^{2}-2\rho \left(\frac{x-\mu_{X}}{\sigma_{X}}\right)\left(\frac{y-\mu_{Y}}{\sigma_{Y}}\right)+{\left(\frac{y-\mu_{Y}}{\sigma_{Y}}\right)}^{2}\right]}$. Subsequently, we must compute the margins of the different code areas. As shown in Figure 3a, the probability density function of $Subject_{a}$ (yellow curve) overlaps the solid line (i.e., the population center). The area of this overlap indicates the amount of error we would suffer for $Subject_{a}$ if the feature is selected for key generation. Specifically, for $Subject_{a}$ whose probability distribution is mostly on the left side of the population center, the feature can be encoded correctly as 0 during key generation in most cases. However, due to the intra-class variation, there could be an error while encoding the feature if it appears on the right side of the population center. In this paper, we assign the maximum allowable overlap as α for a feature. If α is larger, the overlap area will be larger, and the intra-class reliability will be lower. Consequently, α controls the probability of error in (or the reliability of) the key generation, which can be utilized to manage the trade-off between reliability and entropy. For example, in the 1-bit encoding case, only features that satisfy the following constraints can be selected for key generation:(2)$$\int_{\mu_{pop}^{f}}^{+\infty }PDF_{S_{i}}^{f}\leq \alpha ,\;\mathrm{if}\;\mu_{S_{i}}^{f}<\mu_{pop}^{f}$$
(3)$$\int_{-\infty }^{\mu_{pop}^{f}}PDF_{S_{i}}^{f}\leq \alpha ,\;\mathrm{if}\;\mu_{S_{i}}^{f}>\mu_{pop}^{f}$$

With the above restrictions, it is obvious that the margin is directly related to the standard deviation of the probability density function of the population. Thus, it is appropriate to exploit the average standard deviation or the maximum standard deviation of all subjects to compute the suitable margin. In this paper, we adopt the maximum standard deviation for margin computation, which represents the worst-case noise. Specifically, the margin $m^{f}$ can be computed as follows for the 1-bit case:(4)$$\int_{0}^{m^{f}}N\left(0,\sigma_{S_{j}}\right)=\frac{1}{2}-\alpha ,\;\sigma_{S_{j}}=\max_{i=1}^{k}\left\{\sigma_{S_{i}}\right\}$$

In addition, the margins $m_{X}^{f}$ and $m_{Y}^{f}$ for the x-axis and y-axis, respectively, are computed analogously for the 2-bit case as follows:(5)$$\begin{array}{cc} \; & \int_{0}^{m_{Y}^{f}}\int_{0}^{m_{X}^{f}}N\left(\left[0,0\right],\left[\begin{array}{cc} \sigma_{S_{j},X} & 0\\ 0 & \sigma_{S_{j},Y} \end{array}\right]\right)=\frac{1}{2}-\alpha ,\\ \; & \left|\begin{array}{c} \sigma_{S_{j},X}^{2}+\sigma_{S_{j},Y}^{2} \end{array}\right|=\max_{i=1}^{k}\left|\begin{array}{c} \sigma_{S_{i},X}^{2}+\sigma_{S_{i},Y}^{2} \end{array}\right| \end{array}$$

Note that the above equations can also be extended to three or more bits using the proposed approach.

**Individual-Specific Feature Selection and Key Generation.** After determining the population center and margins, it is essential to select reliable features for each subject (the identification results using all features with different classifiers are shown in Appendix A, which are relatively low). First, for subject $S_{i}$, we compute the center point by fitting the probability distribution of $S_{i}$ into Gaussian functions. For feature *f*, we define *f* is reliable for $S_{i}$ when the distance of the center point of $S_{i}$ from the center point of population $\mu_{pop}^{f}$ is greater than the margin $m^{f}$, as shown in (Equation 6) and (Equation 7) for 1-bit and 2-bit cases, respectively:(6)$$|\mu_{S_{i}}^{f}-\mu_{pop}^{f}|\geq m^{f}$$
(7)$$|\mu_{S_{i},X}^{f}-\mu_{pop,X}^{f}|\geq m_{X}^{f},\left|\mu_{S_{i},Y}^{f}-\mu_{pop,Y}^{f}\right|\geq m_{Y}^{f}$$

Then, we compute each sample of the training set of $S_{i}$ to determine whether the feature *f* is reliable. Here, we employ another parameter β for feature selection, and if the proportion of the valid samples in the training set is greater than β, feature *f* is considered to be reliable for $S_{i}$. The indices of features for each subject are stored as helper data. Note that, provided the entropy is sufficiently high, the helper data do not leak essential information.

After selecting reliable features, we can generate keys with particular encoding rules. Figure 3 illustrates the fundamental encoding principle, where each feature is encoded as [0,1] and [00,01,11,10] separately. With the population center and margins, we can compute the boundaries of the code areas for each feature. For example, in the 1-bit case, the left boundary is $\mu_{pop}^{f}-m^{f}$ and the right boundary is $\mu_{pop}^{f}+m^{f}$. The encoding rules for both the 1-bit and 2-bit cases are described as follows:(8)$$code=\left\{\begin{array}{c} 0,\;\mu_{S_{i}}^{f}<\mu_{pop}^{f}-m^{f}\\ 1,\;\mu_{S_{i}}^{f}>\mu_{pop}^{f}+m^{f} \end{array}\right.$$
(9)$$code=\left\{\begin{array}{c} 00,\;\mu_{S_{i},X}^{f}<\mu_{pop,X}^{f}-m_{X}^{f}\;\&\;\mu_{S_{i},Y}^{f}>\mu_{pop,Y}^{f}+m_{Y}^{f}\\ 01,\;\mu_{S_{i},X}^{f}>\mu_{pop,X}^{f}+m_{X}^{f}\;\&\;\mu_{S_{i},Y}^{f}>\mu_{pop,Y}^{f}+m_{Y}^{f}\\ 10,\;\mu_{S_{i},X}^{f}<\mu_{pop,X}^{f}-m_{X}^{f}\;\&\;\mu_{S_{i},Y}^{f}<\mu_{pop,Y}^{f}-m_{Y}^{f}\\ 11,\;\mu_{S_{i},X}^{f}>\mu_{pop,X}^{f}+m_{X}^{f}\;\&\;\mu_{S_{i},Y}^{f}<\mu_{pop,Y}^{f}-m_{Y}^{f} \end{array}\right.$$

Finally, we compute the code of each reliable feature and combine these codes as the target key template for the subject. The key template is then stored in a secure storage medium, e.g., a database on a disk storage device.

**Individual authentication.** In the authentication process, we input the features of the user and combine the codes regenerated with reliable feature indices in the helper data to obtain the key. Here, we utilize parameter γ as the threshold to distinguish the correct subject from other subjects. Specifically, we compute the Hamming distance of the regenerated key with the key template as the key similarity. Each subject has a separate threshold γ, and it is matched only when the key similarity is greater than γ of the target subject. To obtain the optimal γ value for each subject, we utilize a validation set to regenerate keys and compute the evaluation metrics with different γ values. Finally, for a given γ value, we regenerate keys using the test set and obtain the authentication performance.

In this paper, to acquire sufficiently long keys and realize high computational efficiency, we consider the 2-bit case in our experiments, and the pseudo code is shown in Algorithms 1–3. As mentioned previously, each trial has 96 features, which are placed into pairwise combinations to obtain $C_{96}^{2}=4560$ feature combinations in two-dimensional space for key generation.

**Algorithm 1.** MGFBA Population Parameter Computation**Input:** α, population statistics 1: μpop,Xf, μpop,Yf, σpop,Xf, σpop,Yf← Gaussian fitting with population statistics 2: mXf, mYf← Equation (Equation 5) using α, σpop,Xf, σpop,Yf**Output:** population center point (μpop,Xf, μpop,Yf), population standard deviation (σpop,Xf, σpop,Yf), margin(mXf, mYf)


**Algorithm 2.** MGFBA Feature Selection and Key Generation**Input:** β, individual statistics, population center point, margin
 1: Initialization: i ← 1, N ← number of individuals 2: **for** i ← 1 N **do** 3:   $\mu_{S_{i},X}^{f}$, $\mu_{S_{i},Y}^{f}$← Gaussian fitting with individual *i* statistics 4:   $f_{i,1}$, $f_{i,2},...$← Equation (Equation 7) using β, $\mu_{pop,X}^{f}$, $\mu_{pop,Y}^{f}$, $m_{X}^{f}$, $m_{Y}^{f}$ 5:   $\gamma_{i}$, $k_{T,i}$← Equation (Equation 9) using $f_{i,1}$, $f_{i,2}$, $\mu_{pop,X}^{f}$, $\mu_{pop,Y}^{f}$, $m_{X}^{f}$, $m_{Y}^{f}$, $\mu_{S_{i},X}^{f}$, $\mu_{S_{i},Y}^{f}$ 6: **end for****Output:**
reliable feature index of individual *i*($f_{i,1}$, $f_{i,2},...$), similarity threshold of individual *i*($\gamma_{i}$), key template of individual *i*($k_{T,i}$)


**Algorithm 3.** MGFBA Individual Authentication**Input:** γ, individual statistics, population center point, margin, reliable feature index of individuals, similarity threshold of individuals, key template of individuals
 1: Initialization: i ← 1, N ← number of individuals 2: **for** i ← 1 N **do** 3:   $\mu_{S_{i},X}^{\prime f}$, $\mu_{S_{i},Y}^{\prime f}$← Gaussian fitting with individual *i* statistics 4:   $k_{i}$← Equation (Equation 9) using $f_{i,1}$, $f_{i,2}$, $\mu_{pop,X}^{f}$, $\mu_{pop,Y}^{f}$, $m_{X}^{f}$, $m_{Y}^{f}$, $\mu_{S_{i},X}^{\prime f}$, $\mu_{S_{i},Y}^{\prime f}$ 5:   $Res_{i}$← compare $k_{i}$ and $k_{T,i}$ using $\gamma_{i}$ 6: **end for****Output:**
Authentication result of individual *i*($Res_{i}$)


## 3. Experimental Setup and Results

### 3.1. Data Acquisition

The main dataset used to evaluate the proposed MGFBA method included LFP signals recorded in rats. For these animal experiments, 10 adult male Sprague-Dawley rats (300–350 g) were used.

**Behavior tasks.** To validate the cancellability of MGFBA, the rats were trained to perform three unique behavior tasks: (1) running on a treadmill, (2) single-pellet retrieval, and (3) lever pressing. Here, each rat followed a training process for approximately one week. **Treadmill running.** A 80 cm × 9 cm × 12 cm treadmill was utilized to encourage the rats to run. Here, the speed was set to 10 m/min. For each experiment day, we collected five minutes of running data for each rat. The data were inspected visually to remove periods in which the rats were not running. **Single-pellet retrieval.** We used a reaching chamber where pellets were placed in front of a hole on the chamber for the rats to grab. Here, the rat stretching out and drawing back its paw was considered a single trial, and a laser facility was used to record the timestamp of each trial. We trained the rats to perform this task with the left forelimb. During this experiment, the reaching timestamps were recorded, and for each session, the rats performed the grabbing task for 10 min. **Lever pressing.** Here, we used a behavior box, and we trained the rats to perform a lever press task, where water was provided as a reward when the rats pressed the lever down over a pressure threshold. All rats were required to use the left forelimb to perform this task. Prior to each experiment day, water was restricted moderately to 10–15 mL per day to enhance the rats’ lever-pressing performance. During the experiment, the pressure values of the lever were recorded for subsequent analysis. In each session, the rats were required to press the lever for a total of 15 min.

**Neural signal collection and dataset partition.** The LFP signals were recorded using a 16-channel (2×8) handmade microelectrode array (35 μm nichrome) implanted over the right premotor cortex (RFA; 2.5 mm lateral and 3.5 mm anterior to bregma), as shown in Figure 4. The anterior 2×4 electrodes were implanted in the RFA, and the posterior 2×4 electrodes were in the CFA at a depth of 1.2–1.5 mm in layer V. All data were recorded using a multichannel neural signal acquisition system (OmniPlex/128, Plexon TM, Dallas, TX, USA) with an amplification of 1750 and 50-Hz notch filter. The behavior data were recorded simultaneously with neural signals at a sampling rate of 1 kHz. For different behavior tasks, we collected the neural and behavior data for 5∼11 days. We utilized 80% of the dataset as a training set, 10% as a validation set, and 10% as a test set. We employed the training set to compute the population parameters, select reliable features, and generate key templates for subjects, and we used the validation set to select the optimal values for parameters α, β, and γ.

### 3.2. Evaluation Metrics

Our primary focus is identity authentication; thus, we considered common metrics used to evaluate biometric authentication, such as authentication accuracy, false acceptance rate, and key length. The formulas of the authentication accuracy and false acceptance rate are as follows:(10)$$Authentication\;Accuracy=\frac{1}{N}\left(\sum_{i=1}^{N}isValid\left(k_{i},k_{T}\right)\right)$$
(11)$$False\;Acceptance\;Rate=\frac{\sum_{i=1}^{M}\sum_{j=1}^{N_{i}}isValid\left(k_{ij},k_{T}\right)}{\sum_{i=1}^{M}N_{i}}$$
(12)$$isValid\left(k_{1},k_{2}\right)=\left\{\begin{array}{c} 1,\;HD(k_{1},k_{2})\leq \gamma \\ 0,\;HD(k_{1},k_{2})>\gamma \end{array}\right.$$

For a subject *S*, we calculated the authentication accuracy by counting the number of valid keys using all samples of subject *S* (one sample generated one key). Here, we compared the value of HD (Hamming Distance) between the generated key $k_{i}$ and the key template $k_{T}$ of subject *S* to validate the generated key as shown in Equation (Equation 12), where γ is a similarity threshold. As for the false acceptance rate, we computed the number of valid keys using all samples of other subjects, where *M* represents the number of other subjects and $N_{i}$ represents the number of samples of other subject *i*.

In addition, to measure the randomness of the generated keys, we computed the entropy of the keys, which should be sufficiently large to resist attacks. Here, we calculated the min-entropy of feature *f* as follows:(13)$$E\left(f\right)=-\delta log_{2}\left(\max_{i=1}^{n}\left\{P_{i}\left(f\right)\right\}\right)$$
(14)$$P_{i}\left(f\right)=\frac{Count(X=B_{i})}{\sum_{j=1}^{n}Count\left(X=B_{j}\right)}$$
where $B_{i}$ represents the $i_{th}$ bit combination in the total *n* bit combinations, i.e., $B_{i}\in \left\{0,1\right\}$(n=2) and $B_{i}\in \left\{00,01,11,10\right\}$(n=4) for the 1-bit and 2-bit cases, respectively. Here, Count(X) is the number of samples in code area *X*, $P_{i}\left(f\right)$ represents the probability of bit combination *i*, and E(f) is the proportion of the bit combination with the maximum samples. In addition, δ is a normalizing parameter, which is 1 and $\frac{1}{2}$ for the 1-bit and 2-bit cases, respectively. Note that the maximum value of min-entropy is 1 if and only if $P_{i}\left(f\right)=\frac{1}{2^{m}}\left(\forall i\in m,m=1,2,...\right)$, where *m* is the length of the bit combination. If the min-entropy is close to 1, it is nearly impossible to guess the correct bit combination for the feature. In our evaluations, we computed the average min-entropy of all features to investigate the security performance of the proposed method.

### 3.3. Performance of Brain-Based Authentication

Table 1 shows the evaluation results obtained for the three behaviors for all rats. For the running behavior, seven rats achieved 100% authentication accuracy, six rats obtained a 0% false acceptance rate, and the average authentication accuracy and false acceptance rate were 87.5% and 3.9%, respectively. For the grabbing behavior, eight rats obtained 100% authentication accuracy, seven rats obtained a 0% false acceptance rate, and the average authentication accuracy and false acceptance rate were 84.0% and 1.8%, respectively. In terms of the pressing behavior, eight rats obtained 100% authentication accuracy, a total of 10 obtained a 0% false acceptance rate, and the average authentication accuracy and false acceptance rate were 92.9% and 0%, respectively. These results demonstrate that that, among the three behaviors, the best performance was obtained for the pressing behavior. The difference in performance between the behaviors may be due to the fact that the grabbing behavior always involved a chewing noise from the rats, and the running behavior was strenuous with more electromyography and background noise. In contrast, the pressing behavior was conducted in an insulated box, and this behavior is more moderate than the other behaviors and involves less noise-related influence. In addition, the average key lengths were 892 bits, 1209 bits, and 713 bits for the running, grabbing, and pressing behaviors, respectively. These key lengths are all sufficiently long to satisfy security requirements. Note that some rats obtained considerable key lengths. For example, as shown in Table 1, Rat-6 obtained 3780 bits, 4222 bits, and 4412 bits with 100% authentication accuracy and 0% false acceptance rate for the three behaviors, which demonstrates that the proposed method can generate sufficiently long keys with high reliability and security. We also found that rats have their own preferred behaviors. For example, Rat-5 obtained 938 bits for the running behavior while obtaining only 32 bits for the pressing behavior. Similarly, Rat-10 obtained 1636 bits for the grabbing behavior while obtaining only 408 bits for the pressing behavior. It is likely that different subjects may achieve better performance with specific behaviors, which can inform us in terms of selecting optimal behaviors for different subjects. In addition, as shown in Figure 5a, the average entropy of the total 4560 feature combinations was 0.75, 0.68, and 0.69 for the three behaviors, respectively, which is sufficiently high for key security. Overall, we found that the performance obtained for all behaviors was satisfactory.

### 3.4. Revocability

To verify the revocability of the proposed method, we computed the false acceptance rate of other behaviors relative to a given behavior. As shown in Figure 5b, the false acceptance rate was effectively less than 10% for the grabbing and pressing behaviors, while that of the running behavior was greater than that of other behaviors. Specifically, the false acceptance rates of other behaviors relative to running behavior were 13.0% and 24.1%, those for the grabbing behavior were 11.2% and 9.6%, and those for the pressing behavior were 7.7% and 4.7%. These results are relatively higher than the values of the false acceptance rates shown in Table 1, and we discuss the reasons for these differences in the following. On the one hand, the similarity of the brain signals between the behaviors of a rat is naturally higher than the similarity of a behavior between rats. On the other hand, electrode noise that was not eliminated during the preprocessing step could be present; thus, the brain signals may have some duplication between the behaviors of a rat. For example, the brain signals for the pressing behavior exhibited the minimum noise among the three behaviors; therefore, the duplication portion of noise is smaller, and the performance is better. However, for the running behavior, due to the strenuous movements involved, there were more electromyography and background noise, which reduced the proportion of the effective brain signals. If we further optimize the preprocessing steps to reduce additional noise, we think that the false acceptance rate can be controlled within 5%. Overall, we think that it is feasible to generate different reliable keys between behaviors with the proposed method.

**Figure 5 bioengineering-10-00912-f005:**
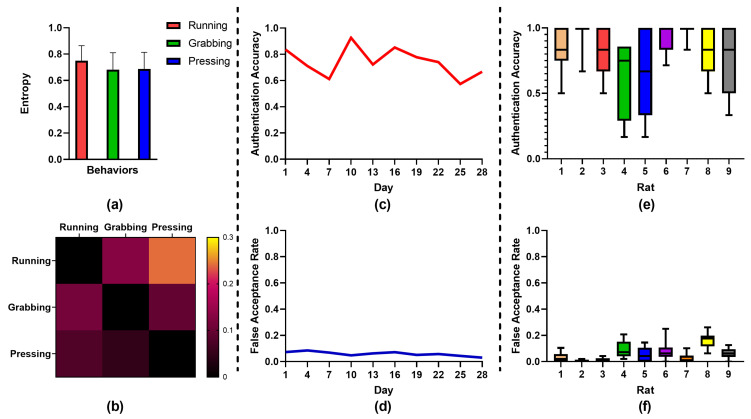
(**a**) Average entropy of three behaviors. (**b**) Performance of revocability between behaviors. (**c**) Average authentication accuracy of nine rats over 28 days. (**d**) Average false acceptance rate of nine rats over 28 days. (**e**) Average authentication accuracy of each rat over 28 days. Each color corresponds to a rat. (**f**) Average false acceptance rate of each rat over 28 days. Each color corresponds to a rat.

### 3.5. Long-Term Stability

To examine the long-term stability of the proposed method, we collected the brain signals for the ruxnning behavior every three days for 40 days with nine rats (one rat died prior to the acquisition). Specifically, in total, we obtained 14-day data after one month of collection. Here, we utilized previous 4-day data for training and the remaining 10-day data for test (the duration of test is 28 days). As shown in Figure 5c, the authentication accuracy decreased slightly over time, and the authentication accuracies of test day-1 and test day-28 were 83.6% and 66.7%, respectively. As shown in Figure 5d, the value for the false acceptance rate also declined. The false acceptance rates of test day-1 and test day-28 were 7.3% and 3.0%, respectively. The reason for the above results may be that the noise in the brain signals was larger due to electrode drift. With more noise signals and less reliable features, the authentication accuracy decreased. However, due to the irregular noise signals, the probability of false authentication was also lower. Surprisingly, some rats exhibited excellent performance during the long-term experiments. As shown in Figure 5e, Rat-2 and Rat-7 effectively obtained nearly 100% authentication accuracy from test day-1 to test day-28, and Rat-1 and Rat-6 achieved greater than 80% authentication accuracy from test day-1 to test day-28. These results demonstrate that the proposed method realizes outstanding long-term stability. However, the authentication accuracy of some rats was relatively low, e.g., Rat-4 and Rat-5. Possible reasons for this poor performance may include obvious electrode drift and biological changes in the brain’s neural cells, thereby leading to increased noise and less effective information for the key generation process. Overall, we achieved an average authentication accuracy of 74.2% and an average false acceptance rate of 5.9% for nine rats during the one-month experimental period, which demonstrates that the proposed method is sufficiently practical for long-term key generation. We believe that the long-term performance can be improved by using superior electrode materials and implementing better preprocessing steps to eliminate more noise.

### 3.6. Influence of Parameters

#### 3.6.1. Training Size

In the following, we examine the influence of different numbers of training days on the key generation performance. Here, we computed the evaluation metrics of training days 1 to 8 for the running and grabbing behaviors and days 1 to 4 for the pressing behavior. As shown in Figure 6a–c, for all three behaviors, the authentication accuracy increased slightly using different numbers of training days, the false acceptance rate was approximately 3%, and the entropy was approximately 0.7. These results prove that the brain signals remained effective over several days, and the performance was stable. In addition, we obtained an authentication accuracy of approximately 80% using only one training day. These results demonstrate that we can utilize fewer training days to realize comparable performance, which can greatly reduce the pressure for training models and provide convenience in practical application.

#### 3.6.2. Number of LFP Channels

We also investigated the influence of different numbers of training channels on the key generation performance, including 4∼96 channels. As shown in Figure 6d, the authentication accuracy increased rapidly up to 32 channels for both the running and grabbing behaviors and up to 16 channels for the pressing behavior. Then, the authentication accuracy remained stable for all three behaviors with increasing numbers of training channels. Thus, we consider that using 32 channels for the running and grabbing behaviors and 16 channels for the pressing behavior is sufficient for the key generation process. In terms of the false acceptance rate, the values decreased slightly with an increasing numbers of channels. It is likely that with more channels, we can obtain more essential differences of the brain signals between rats, which is expected to enhance the invasion difficulty. For entropy, we found that the values increased up to 16 channels and then remained stable. A possible reason for this increase is that the population information was inadequate for stable feature distribution with fewer than 16 channels. Overall, it is feasible that we can adopt fewer channels to generate stable and secure keys, which can ease the demand for the electrode and acquisition equipment.

#### 3.6.3. Parameter α

In this section, we compare the performance of the three behaviors with different α values ranging from 0.20 to 0.49. As shown in Figure 7, the authentication accuracy increased with increasing α values. We consider the reason could be that there may be few stable combinations with a low α value, which has stricter restrictions in terms of key generation, and this leads to low authentication accuracy. In terms of the false acceptance rate, the value increased slightly with the increasing α value, which is reasonable due to the narrower boundaries and larger code areas. In addition, for both entropy and key length, more reliable feature combinations are joined; thus, the results are improved with larger α values.

#### 3.6.4. Parameter β

We also observed the performance of the three behaviors using different β values ranging from 0.1 to 1.0. As shown in Figure 8, the authentication accuracy increased up to a β value of 0.9 and then declined. Obviously, with a higher β value, we could pick out more stable combinations, which could generate more reliable keys and improve authentication accuracy. However, when β reached 1.0, the restriction was too high to regenerate identical keys; thus, the accuracy declined considerably. In terms of the false acceptance rate and key length, the values increased slightly due to the lower number of reliable combinations. While entropy is not associated with β, the value was constant using different β values.

## 4. Discussion

To the best of our knowledge, this study is the first to propose a brain-based key generation method that uses intracortical brain signals. In this paper, we have demonstrated the potential to provide both reliable and revocable biometrics. We found that the proposed MGFBA method can reliably generate long keys (up to 4412 bits) from the LFPs, which is suitable for various security applications, e.g., electronic cash applications and secure multiparty computation. In addition, the proposed brain biometrics can be revoked easily by changing the paradigms. However, in this study, we only experimented on rats with three specific behaviors; thus, the theoretical challenge and response pairs may be huge. In addition, we found that the proposed method also obtains good long-term stability over a one-month period, which demonstrates its feasibility for practical use.

Currently, most studies utilize EEG as the main source for biometric construction due to the convenience of signal acquisition [19,20,21,22,23,24]. However, a major problem with EEG is its low SNR because the brain’s electrical signals decay significantly while passing through the skull and scalp. Thus, an EEG-based system cannot realize sufficiently high reliability due to the signal quality, and the revocability is highly limited [25]. In addition, the electrode drift problem with EEG is an inherent limitation once the EEG cap is removed and then worn again, which results in insufficient long-term stability. A previous study [26] evaluated the performance of EEG over time and demonstrated a significantly decreasing trend. Thus, using intracortical brain signals, e.g., LFPs, can provide a better option for high-performance brain biometric construction because these signals are recorded with electrodes placed directly on the cortex such that they are higher resolution with sufficient SNR [27,28].

We also identified several limitations with the proposed method. Notwithstanding, brain biometrics also exist many potential problems. In this study, we utilized intracortical brain signals to obtain sufficient performance; however, the implanted electrodes used for signal acquisition may be invalid or suffer from the drift problem, which would lead to a reduction in long-term stability. In addition, the noise of the implanted electrodes may mix into the effective brain signals; thus, such noise may be incorrectly considered as effective signals. However, we believe that these limitations can be overcome with the ongoing development of acquisition equipment and optimization of the preprocessing steps. As shown in Figure 4e,f, the long-term performance obtained with Rat-2 and Rat-7 was excellent, with nearly 100% authentication accuracy and a 0% false acceptance rate, which demonstrates that brain signals are capable of providing long-term stability. Thus, with more stable equipment, the performance of brain biometrics can be improved further.

Similarly, the cancellability performance can also be affected by the noise of implanted electrodes. Such noise may be computed as features for key generation, which could increase the similarity of keys between different behaviors. We also found that the intensity of the behaviors influences the false acceptance rate between behaviors, as shown in Figure 4b. With increasing intensity, the noise of the electrode becomes obvious. With slight behaviors, e.g., the pressing behavior, the performance was significantly better than that for the running behavior. Thus, electrode noise is the main factor influencing cancellability performance, which could be solved by further optimization of the preprocessing steps. Overall, our results prove that the cancellability of brain metrics is good, and the performance can be improved further with additional optimization.

## 5. Conclusions

In this paper, we have proposed a key generation method that extracts keys from the LFP with high reliability and entropy. In addition, we found that intracranial brain signals can be excellent biometric features with outstanding reliability and cancellability due to the higher spatial and temporal resolution than EEG signals. The proposed MGFBA approach can reliably generate long keys (up to 4412 bits) from the LFPs, which is suitable for various security applications, e.g., electronic cash applications and secure multiparty computation. The experimental results demonstrate that a 938-bit key on average can be generated from the LFP signals from three behaviors at an authentication accuracy of 88.1% and false acceptance rate of 1.9%. Furthermore, we can easily transform the behaviors to generate different reliable and secure keys, which proves that brain biometrics are revocable. We think that our work has demonstrated the potential of using intracortical brain signals for reliable authentication and other security applications, e.g., brain–computer interfaces.

## Figures and Tables

**Figure 1 bioengineering-10-00912-f001:**
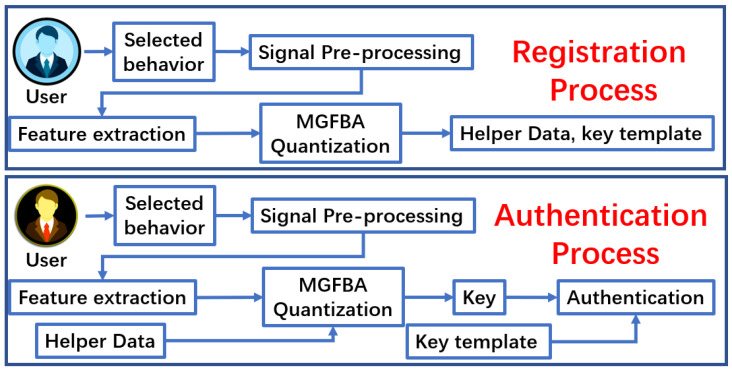
Flowchart of registration and authentication of the biometrics.

**Figure 2 bioengineering-10-00912-f002:**
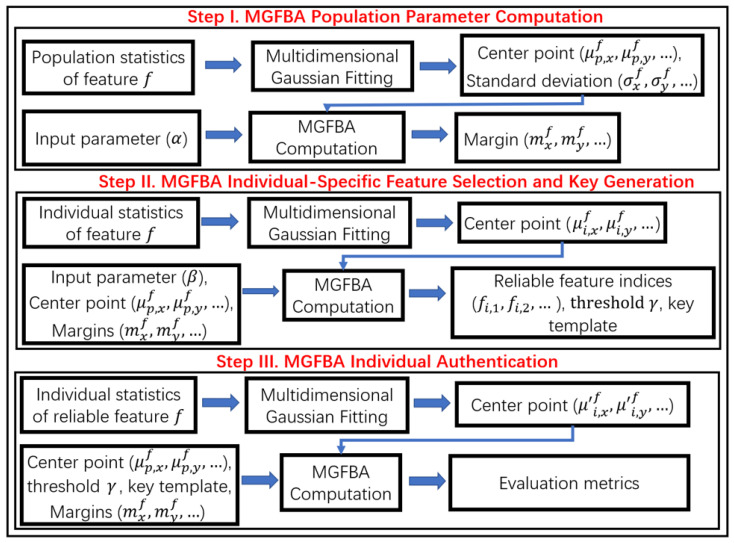
Block diagrams of MGFBA quantization.

**Figure 3 bioengineering-10-00912-f003:**
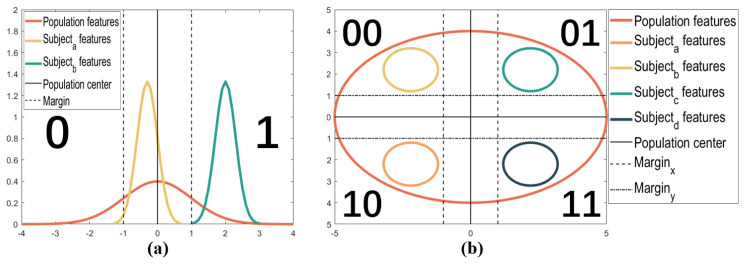
Feature encoding and quantization of the (**a**) 1-bit case and (**b**) 2-bit case with the proposed MGFBA method. For the 1-bit case, the central red curve represents the probability distribution of the entire population, and the other two curves represent the probability distributions of different subjects in the population. For the 2-bit case, the central red circle represents the probability distribution of the entire population, and the other four circles represent the probability distributions of different subjects in the population. The solid and dotted lines represent the population center and the boundaries of different code areas, respectively.

**Figure 4 bioengineering-10-00912-f004:**
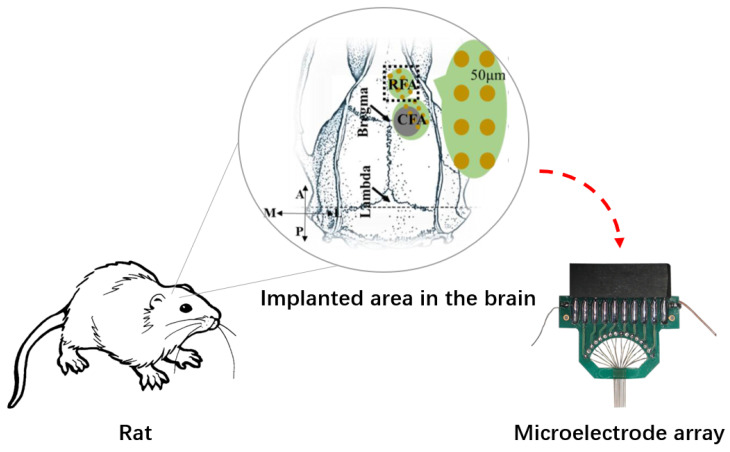
Schematic diagram of intracortical brain signal collection of the rat. The gray circle of CFA stands for the ischemic area, and the green areas of both CFA and RFA show the location of electrodes (dark yellow). A 16-channel (2×8) handmade microelectrode array (35 μm nichrome) was implanted to collect the brain signals.

**Figure 6 bioengineering-10-00912-f006:**
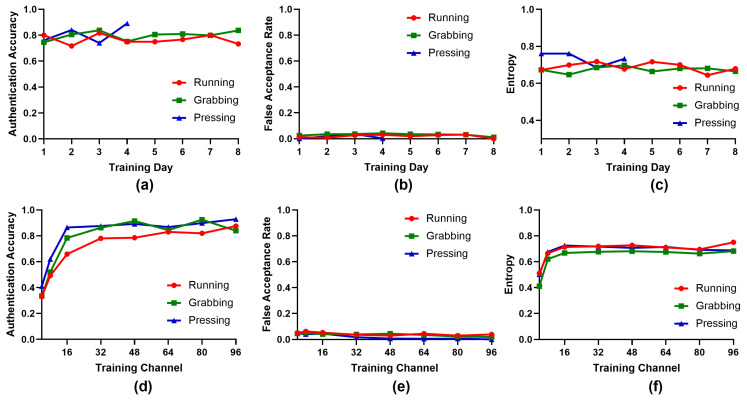
Performance using different numbers of training days and training channels. (**a**) Average authentication accuracy using different numbers of training days. (**b**) Average false acceptance rate using different numbers of training days. (**c**) Average entropy using different numbers of training days. (**d**) Average authentication accuracy using different numbers of training channels. (**e**) Average false acceptance rate using different numbers of training channels. (**f**) Average entropy using different numbers of training channels.

**Figure 7 bioengineering-10-00912-f007:**
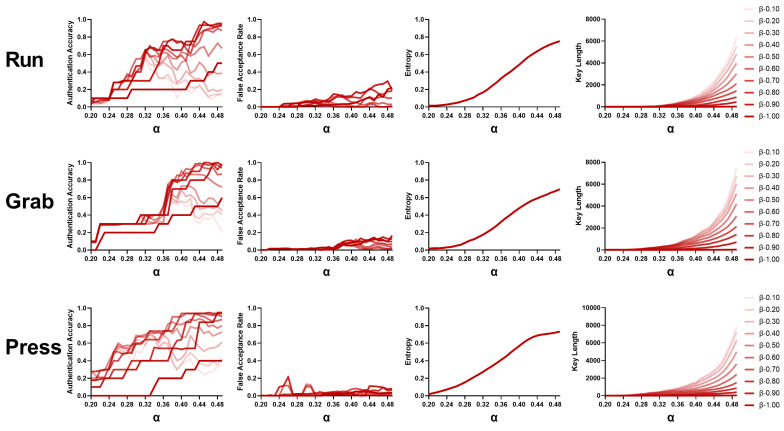
Performance obtained with different α values (0.20–0.49) for three behaviors.

**Figure 8 bioengineering-10-00912-f008:**
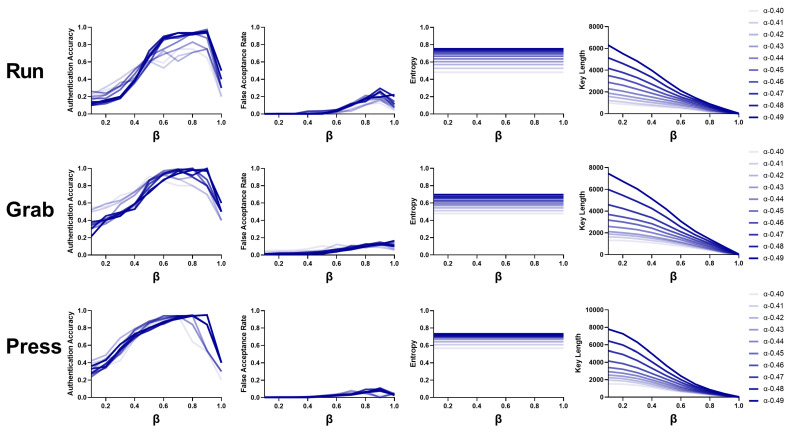
Performance obtained with different β values (0.1–1.0) for three behaviors.

**Table 1 bioengineering-10-00912-t001:** Performance with different motor behaviors.

**RUNNING**	Rat-1	Rat-2	Rat-3	Rat-4	Rat-5	Rat-6	Rat-7	Rat-8	Rat-9	Rat-10	**Avg**
Authentication Accuracy	1.00	0.75	0.50	0.50	1.00	1.00	1.00	1.00	1.00	1.00	**0.88**
False Acceptance Rate	0.08	0	0	0.11	0	0	0.11	0	0.08	0	**0.04**
Key Length	1114	338	488	248	938	3780	740	112	30	1134	**892**
**GRABBING**	Rat-1	Rat-2	Rat-3	Rat-4	Rat-5	Rat-6	Rat-7	Rat-8	Rat-9	Rat-10	**Avg**
Authentication Accuracy	0.23	1.00	0.17	1.00	1.00	1.00	1.00	1.00	1.00	1.00	**0.84**
False Acceptance Rate	0	0	0	0	0	0	0.11	0.05	0.03	0	**0.02**
Key Length	2110	134	140	358	250	4222	2858	234	148	1636	**1209**
**PRESSING**	Rat-1	Rat-2	Rat-3	Rat-4	Rat-5	Rat-6	Rat-7	Rat-8	Rat-9	Rat-10	**Avg**
Authentication Accuracy	1.00	1.00	1.00	1.00	1.00	1.00	0.89	0.40	1.00	1.00	**0.93**
False Acceptance Rate	0	0	0	0	0	0	0	0	0	0	**0**
Key Length	642	48	360	150	32	4412	650	366	60	408	**713**

## Data Availability

The datasets are available on reasonable requests from the corresponding author.

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
