# Peer review of "Encrypt with Your Mind: Reliable and Revocable Brain Biometrics via Multidimensional Gaussian Fitted Bit Allocation"

_bioengineering, 2023, doi:10.3390/bioengineering10080912_

Round 1

Reviewer 1 Report

This study presented a novel brain-based key generation approach called multidimensional Gaussian fitted bit allocation (MGFBA). The presented MGFBA-based keys can be revoked using different motor behaviors. The authors conducted experiments to demonstrate the potential of using intracortical brain signals for reliable authentication. Although this study provides contributions in biometric-based authentication,

There are few issues needed to be improved and answered before publication.

1) Figure 3 necessarily larger letter size (in other graphs it would also make it easier to read). 

2) The authors should provide formulas of accuracy and false acceptance rate.

3) Is it possible to compare performance between the proposed biometric authentication and other previous work?

Reviewer 2 Report

The idea of the work is good and interesting. However, I would not agree with some aspects or ideas. 

1. The authors write: "..most biometrics are not cancellable .....". There is a biometrics field called 'Cancellable Biometrics' and there are works and patents on that. They show how we can change the pattern of our fingerprints, for example. I suggest the authors should study that and consider some example in their work.

2. Biometrics is not the plural of 'biometric' ! Biometrics IS (not ARE). To talk about biometrics trait, we say biometrics feature. Hence, the authors should make the necessary changes in the whole paper.

3. I could not imagine how the authors collect the necessary signals from the brain - how and where they put the necessary probes for measurements. A figure of illustration would be sufficient.

4. The flow charts suggest good description but are not covering the computer program implementation of the described algorithms. A pseudocode will be of great help to the computer engineering experts.

5. There are more than one conclusion, hence section 5. should be CONCLUSIONS.  

As I mentioned above, biometrics is singular (like news). Hence, through the whole work the word biometric should be changed in 'biometrics feature'. And 'biometrics are' should be changed into '... is'.

Reviewer 3 Report

The authors proposed a Multidimensional Gaussian Fitted Bit Allocation (MGFBA) method to identify 7 rats performing three behavior tasks using LFP features. The proposed method showed good identification results. However, there are a few questions the authors did not stress, albeit they are relevant.

1.     The authors extracted different frequency band features as mentioned in line 110. The reasons for choosing these bands are not given in the manuscript. Why did the authors skip 4-7 Hz, 20-70 Hz, and 115-130 Hz?

2.     The 96 features used are frequency band power and the sum of spectrum power. The way authors used to calculate this power should be mentioned in the manuscript. Is the power calculated as the amplitude of the corresponding FFT bin or using time domain squared power? Is the spectrum power in dB or not?

3.     The two assumptions assume the calculated features can be seen as multivariate Gaussian distributions for both population and individual. However, all features calculated in 2 are power-related features, which are non-Gaussian by nature. Is the assumption appropriate? Have the authors performed any statistical tests to verify the assumptions?  

4.     The authors used multivariate Gaussian distribution to fit the features, what is the goodness-of-fit?

Apart from the above-mentioned questions, the following recommendations might be helpful.

1.     The authors should consider rearranging their sections. It is only in the section on Experimental setup and results that it becomes clear what are those behavioral tasks and that the "subjects" in this study were rats. This causes some confusion as in the abstract and introduction, this information is not provided.

2.     It would be interesting to look at the robustness of the proposed system by having some unseen imposter rats, which are not present in the population parameter computation stage, and act as negative samples in the individual authentication step. It would be more convincing if the system can classify these unseen examples correctly.

Round 2

Reviewer 3 Report

The authors have addressed my questions in the revised manuscript except for the following:

    1. The log-power features are not normal distributed as they are skewed because they contain positive values only. Thus, assumption 1 of the paper should be modified.

    2. Considering the good results obtained using the proposed method, it is possible that in the high dimensional feature space the multivariate distribution is somehow pseudo-Gaussian as the skewness becomes less obvious when the dimensionality increases. The exact distribution of these features seems to be an important aspect to fully understand how the model works but is not addressed in the manuscript (see also my previous remark). 

    3. Did the authors consider using a simple linear baseline classifier as a reference? The result of reaching a high accuracy might be simply because these features are highly differentiable, given that the authors claimed intracortical signals exhibit good signal quality. Thus, to show that the model actually works, it would be interesting to use the same features and train a simple linear model.

Regarding the method, I still have one question. In section 3.1, Neural signal collection and dataset partition, the authors mentioned the use of a 80-10-10 split for the training, validation set and test set, which does not seem appropriate in view of the long-term stability experiment (section 3.5), if the 80% training data are randomly drawn from the 1-month pool. The long-term stability test should be assessed with another dataset partition strategy.

Round 3

Reviewer 3 Report

The authors have addressed the questions of the 2nd review round. Below my response to the authors’ replies 1 and 2.

In principle, log values can be negative, and can be normal distribution, if the original data is of log-normal distribution. According to your feature distribution plots in the supplementary materials (S1-S10), there are no negative values present, and many of them have a long tail. In fact, the point is not to prove how the distribution exactly looks like. It is more aboutverifying whether your assumptions are correct and gain more insights into the data and the proposed model.  That is the reason why I would like to know the goodness-of-fit of assumptions 1 and 2. With the newly provided supplementary material, it seems the 1-bit and 2-bit examples of Figure 3 do not represent your actual data, as most features from S1-S10 have mean values larger than 10, and no negative values. It is thus recommended to either use your real feature data from S1-S10 to generate this Figure or to mention it is a high-level illustration.